# Implementation measurement in global mental health: Results from a modified Delphi panel and investigator survey

Christopher G. Kemp[1] , Kristen Danforth[2], Luke Aldridge[3], Laura K. Murray[3] and Emily E. Haroz[1,3]

[1]Department of International Health, Johns Hopkins University, Baltimore, MD, USA; [2]University of Washington, Seattle, WA, USA and [3]Department of Mental Health, Johns Hopkins University, Baltimore, MD, USA

## Research Article

global mental health; implementation measurement; Delphi panel; implementation science

**Corresponding author:**
Christopher G. Kemp;
Email: ckemp11@jhu.edu

## Abstract

Limited guidance exists to support investigators in the choice, adaptation, validation and use of implementation measures for global mental health implementation research. Our objectives were to develop consensus on best practices for implementation measurement and identify strengths and opportunities in current practice. We convened seven expert panelists. Participants rated approaches to measure adaptation and validation according to appropriateness and feasibility. Follow-up interviews were conducted and a group discussion was held. We then surveyed investigators who have used quantitative implementation measures in global mental health implementation research. Participants described their use of implementation measures, including approaches to adaptation and validation, alongside challenges and opportunities. Panelists agreed that investigators could rely on evidence of a measure's validity, reliability and dimensionality from similar contexts. Panelists did not reach consensus on whether to establish the pragmatic qualities of measures in novel settings. Survey respondents ($n = 28$) most commonly reported using the Consolidated Framework for Implementation Research Inner Setting Measures ($n = 9$) and the Program Assessment Sustainability Tool ($n = 5$). All reported adapting measures to their settings; only two reported validating their measures. These results will support guidance for implementation measurement in support of mental health services in diverse global settings.

## Impact statement

Growth in the need for rigorous implementation science in global mental health research has outpaced the development and validation of pragmatic tools to measure implementation processes and outcomes in diverse global settings. Of the few implementation measures that are currently in use, essentially all were developed for use in high-income settings, and few have been psychometrically assessed or validated. Our objectives were to (1) bring together a panel of experts and build consensus around best practices for implementation measurement in diverse global settings and (2) survey investigators applying these measures to identify strengths and opportunities in current practice. The results will support guidance for use by investigators planning to quantitatively measure implementation process and outcomes in diverse global settings. This guidance could facilitate novel, rigorous and replicable implementation research in areas of high need.

## Introduction

Mental, neurological and substance-use (MNS) disorders are the leading causes of disability globally, yet most people in need of treatment for MNS disorders never receive care (Thornicroft et al., 2017; Pathare et al., 2018; Vos et al., 2020). Effective, affordable, scalable and sustainable services are needed to bridge this global gap (Lancet Global Mental Health Group et al., 2007). A broad range of preventive and treatment interventions for high-burden MNS conditions have demonstrated promising cost-effectiveness in both high- and low-resource settings (Patel et al., 2016); in response, researchers and funders alike have called for an increased scientific focus on strengthening intervention implementation and scale-up, particularly in low- and middle-income countries (LMICs), through the application of the methods of implementation science (Betancourt and Chambers, 2016). The primary aim of implementation science is to design and test ways to promote and sustain the delivery of evidence-based practices in routine healthcare (Eccles and Mittman, 2006). These *implementation strategies* target specific aspects of the environment of service delivery, or of the intervention providers or of the intervention itself, all with the goal of improving uptake and sustainment. Implementation success is assessed through a

range of implementation outcomes, including acceptability, adoption, appropriateness, cost, feasibility, fidelity, penetration and sustainability (Proctor et al., 2011). For example, if unhelpful attitudes or beliefs among clinic staff are thought to be hindering implementation of evidence-based mental health care, the use of peer influencers or opinion leaders might be considered as an implementation strategy to improve provider acceptance of mental health services. Application of implementation science methods to the field of global mental health has grown rapidly in recent years (Wagenaar et al., 2020).

This growth has outpaced the development and validation of pragmatic tools for implementation measurement in diverse global settings. As with any science, valid measurement is critical to the utility and reproducibility of implementation research (Lewis et al., 2015). For example, many implementation studies begin with an assessment of the multi-level contextual determinants of implementation effectiveness (Damschroder et al., 2009). These determinants can inform the choice of implementation strategies; they are also useful for understanding the process of implementation and they may moderate or mediate intervention effects (Waltz et al., 2019). Measurement of implementation outcomes is also critical to judging the effectiveness of implementation strategies. While some implementation constructs may be *manifest*, or measured through observable indicators (e.g., rate of provider serviced delivery as an indicator of penetration) (Willmeroth et al., 2019), many are *latent*, implying some level of self-report (e.g., provider acceptability). Many quantitative measures of latent implementation constructs exist and have been identified and catalogued through systematic review; relatively few, however, have been assessed for validity or have documented strong psychometric properties, though the number of measures with strong psychometric properties is increasing (Khadjesari et al., 2020; Mettert et al., 2020). Even fewer measures have been assessed for their pragmatic qualities, including burden, length, reliability and sensitivity to change (Hull et al., 2022). Importantly, almost all extant, validated, pragmatic, quantitative implementation measures were developed for use in high-income countries (Lewis et al., 2015). These implementation measures – and their corresponding theories, models and frameworks – may need to be appropriately translated, adapted and validated for use in diverse global contexts (Means et al., 2020).

To date, most implementation studies by global mental health researchers have relied exclusively on qualitative assessment, with relatively few using quantitative implementation measures (Wagenaar et al., 2020). Though qualitative methods are a crucial part of implementation science, valid quantitative measurement allows for larger studies and improves study rigor and reproducibility (Palinkas et al., 2011; Palinkas, 2014). Investigators have several factors to consider when choosing quantitative measures for use – in addition to whether an appropriate measure exists – including different aspects of measure validity and reliability, as well as each measure's pragmatic qualities (e.g., length, cost) (Powell et al., 2017). Given that almost all existing implementation measures were developed for use in high-resource settings, global mental health researchers must carefully consider the validity and appropriateness of each measure in their setting. There are several distinct approaches available for establishing validity and other measure characteristics in novel settings (Boateng et al., 2018). Table 1 describes these characteristics and approaches in detail and notes which approaches are designed to assess which characteristics. For example, cross-cultural validity can be established using translation, back-translation, expert advice and pre-testing.

Limited guidance exists to support global mental health services investigators in the choice and use of quantitative implementation

measures – or the choice and use of approaches to adapt and validate those measures. Our objectives in this project were to (1) bring together a panel of experts to better understand and develop consensus on best practices for implementation measurement, with a particular focus on mental health implementation research in LMICs, and (2) survey investigators applying these measures to identify strengths and opportunities in current practice.

## Methods

### *Expert panel*

#### *Participants*

We used purposive sampling to select and invite a panel of experts at the intersection of implementation science, psychometrics and global mental health, starting from a list generated by members of the study team. Specifically, we approached experts in our extended professional networks who we knew had experience with developing, adapting or validating implementation measures for use in global mental health research. We recruited eight panel members (see Supplementary Material for a full list of panel participants). One panel member withdrew between the first and second panel discussions.

#### *Delphi process*

The goal of our modified Delphi process was to develop consensus among the panel members on: (1) prioritization of different types of measure validity, reliability and pragmatic qualities for assessment and confirmation when using measures under different circumstances and in different settings (see Table 1 for definitions of each quality); (2) feasibility and utility of different measure validation approaches (see Table 1 for definitions of each approach) and (3) a minimal set of validation approaches for use when applying implementation measures in new contexts and settings. We followed the steps of a conventional Delphi process, including an exploratory phase, a first round of quantitative questionnaires, analysis/summation and results discussion (Avella, 2016). A preliminary discussion was held in March 2020 to orient panelists to the Delphi process. Questionnaires were then distributed and completed electronically. Questionnaire responses were aggregated and anonymized, and summary statistics of responses were presented to the panel. Following the distribution of the questionnaire analysis, available panel members were convened virtually to review the results and, if possible, achieve consensus on recommendations.

Questionnaires included three sections (see Supplementary Material). In the first section, panel members were given different measurement scenarios (e.g., use of an implementation measure developed in a US context to assess the same construct in a novel, lower-resource context) and were asked which types of measurement characteristics (e.g., different types of validity, reliability or pragmatic qualities; Table 1) need to be established prior to measure use in a novel context. In the second section, panel members rated distinct validation strategies (e.g., informal expert elicitation, pilot survey with subsequent real-world outcomes; Table 1) on nine dimensions of rigor, feasibility and resource intensiveness. Finally, in the third section panel members proposed a minimal set of validation strategies that researchers could use under most circumstances when applying an implementation measure in a diverse new setting.

One author (KD) had access to the questionnaire responses and interview data and completed all analyses (Linstone and Turoff, 1975). To maintain confidentiality and promote the rigor of the

**Table 1.** Implementation measure characteristics mapped to measure assessment approaches

| Study Designs | Summary | Substantive content validity | Discriminant content validity | Predictive validity | Concurrent validity | Convergent validity | Discriminant validity | Know-groups differentiation | Correlation analysis | Cross-cultural validity | Dimensionality | Internal consistency | Test-retest reliability | Acceptable (Stakeholder) | Offers relative advantage over existing methods (Easy) | Completed with ease | Appropriate (Compatible) | Fits organizational activities (Useful) | Informs clinical or organizational decision-making | Cost (Objective – Acceptable) | Uses accessible language | Assessor burden (training) | Assessor burden (interpretation) | Length |
|---|---|---|---|---|---|---|---|---|---|---|---|---|---|---|---|---|---|---|---|---|---|---|---|---|
| *(Validity → Content validity)* Is the measure reflective of the construct of interest? | | | Is the measure non-reflective of other constructs? | Does the measure predict future outcomes? | Does the measure correlate with concurrent gold-standard criterion measurements? | Does the measure correlate with other measures of the same construct? | Does the measure correlate with other measures of different constructs? | How does the measure behave across two or more known groups? | How does the measure correlate with existing measures? | Ensuring a tool is adapted for a new context and performs adequately | Does the measure capture the hypothesized structure of the construct? | Do the scale items co-vary relative to their sum score? | Are participant performances repeatable? | Acceptable | Offers relative advantage over existing methods | Completed with ease | Appropriate | Fits organizational activities | Informs clinical or organizational decision-making | Cost | Uses accessible language | Assessor burden (training) | Assessor burden (interpretation) | Length |
| Informal expert elicitation | Ask a few researchers to consider whether one or more measure's items seem representative of each construct's theoretical content. | X | X (if including multiple measures) | | | | | | | | | | | | | | | | | | | | | |
| Formal expert or target population elicitation (e.g. Delphi) | Ask a large group of researchers and practitioners to rate the extent to which items reflect the constructs they were intended to measure. Perform formal quantitative analysis. | X | X (if including multiple measures) | | | | | | | | | | | | | | | | | | | | | |
| Translation, back-translation, expert advice, pre-testing with cross-sectional or repeat-measures survey | Use professional services and expert advice to translate and back-translate measure to/from target language and cultural context, pre-test with target population | | | | | | | | | X | X | X | X (with repeated measurement) | | | | | | | | | | | |
| Survey with sample of target population, using novel measures and established measures of similar or dissimilar constructs | Conduct a cross-sectional or repeat-measures survey with target population, including new measure(s) as well as established measures of similar and/or dissimilar constructs. | | | | X (with gold standard measure) | X (with measure of similar construct) | (with measure of dissimilar construct) | X (pre-stratified target population) | X | | | X | X (with repeated measurement) | | | | | | | | | | |
| Survey with sample of target population, using novel measures and subsequent real-world outcomes (e.g., EBP adoption, patient health) | Conduct a cross-sectional or repeat-measures survey with target population, including new measure(s) and referent EBP at t=0, and observe relevant outcomes at t=1 | | | X | X (with gold standard measure) | X (with measure of similar construct) | (with measure of dissimilar construct) | X (pre-stratified target population) | X (if including other measures) | | | X | X (with repeated measurement) | | | | | | | | | | |
| Vignette-based approach, using novel measures and established measures of similar or dissimilar constructs | Use measures with target population and randomly assigned vignette that systematically varies with respect to construct of interest. | | | | X | X (with measure of similar construct) | (with measure of dissimilar construct) | X (pre-stratified target population) | X | | | X | X (with repeated measurement) | | | | | | | | | | |
| Process evaluation of measure use | e.g., time-motion costing, qualitative study of stakeholder acceptability of measurement use | | | | | | | | | | | | | X | X | X | X | X | X | X | X | X | X | X |

process, no identifying information was shared with other members of the research team or expert panel. Results draw from all questionnaire and interview responses as well as discussion during the second-round call. CK moderated and LA attended, but did not contribute to, both rounds of panel discussion.

The aim was to achieve a reasonable degree of consensus among panel members. No *a priori* target for degree of consensus was set for this study, and a full consensus-based approach was not pursued. This was done for reasons of appropriateness and feasibility; in particular, there are only a small number of experts at the intersection of global mental health and implementation measurement worldwide, and ongoing travel restrictions and social distancing measures related to the COVID-19 pandemic meant in-person consensus-building activities were impossible at the time. Though we did not use a quantitative threshold (e.g., calculating an agreement statistic or a formal vote) to assess consensus, we did bring the expert panel together for a Zoom-based discussion of the summary of their questionnaire results, with a particular focus on areas of divergence. Panel members agreed with the synthesis of results and concluded that the rankings of results within each subsection were acceptable and reflected their judgement.

### Investigator survey

#### Participants

We also conducted a survey of global mental health researchers to understand current practice in implementation measurement. We searched NIH RePORTER and the Grand Challenges Canada website on May 18, 2020, for descriptions of funded implementation research studies related to mental health services in LMIC settings (see Supplementary Material for the NIH RePORTER search strategy). The names and contact information for the lead principal investigator for each study, as well as study descriptions, were abstracted into a sampling frame. One of three authors (C.G.K., K.D., L.A.) screened each study and associated principal investigator for inclusion; studies were excluded if they were not conducted in an LMIC or were not related to mental health. We contacted all remaining principal investigators and invited them to participate in a structured online survey related to the measurement of implementation processes and outcomes in their study. Principal investigators could also nominate a study team member or collaborator – someone who was directly involved in the implementation measurement component of the study – to participate in their place. Between NIH RePORTER, Grand Challenges Canada and this snowball sampling approach, we anticipated reaching most investigators with experience leading formal global mental health implementation research. Contacted investigators were sent a reminder email if they did not initially respond to the online questionnaire within a 2-week period, and a final reminder was sent 2 weeks later. Survey recruitment and data collection occurred from July to November 2020.

#### Survey measures

We designed the survey to assess: (1) the scope and nature of global mental health implementation research conducted by each investigator, (2) the range of implementation process and outcome measures used by investigators across any of their implementation studies and (3) the study setting, population, sample size, types of measure adaptation or validation used if any, assessment of measure performance and any recommendations for measure improvement.

#### Analysis

Categorical responses were summarized using simple descriptive statistics at the level of the respondent. Open-text responses were reviewed for recurring themes or approaches to adaptation and validation.

### Research ethics

The Human Subjects Division of the University of Washington determined that both components of this study qualified for exemption status under 45 CFR 46.101 (b).

## Results

### Expert panel

#### Section 1: Measure characteristics

There was substantial concordance across panel members indicating it was reasonable to rely on evidence of most measure characteristics that had been established in similar contexts (e.g., another low-resource setting) without needing to establish those characteristics in every new setting (Supplementary Material, Section 1). This was true for all types of measure validity, reliability and dimensionality, except for cross-cultural validity (i.e., adequate adaptation for and performance in a new context), which was judged important to be established in each new setting. In contrast, there was limited agreement on the need to establish the pragmatic qualities of measures in each new setting. Though qualities like measure cost, length, ease of completion and assessor burden were judged to be unnecessary to be established in new settings if already established in similar settings, qualities related to how the measure would be used (e.g., whether it would inform decision-making, whether it fit with organizational activities) were felt to be important to establish in each new setting.

Panel members were then asked whether it was ever possible to rely on evidence of measure characteristics that had been established in other settings, even settings that were substantially different (e.g., high-income country). Respondents indicated that if investigators established the face validity of an implementation measure in a new setting – for example, through informal expert review and a small pilot use with confirmatory factor analysis – it would not then be necessary to conduct an intensive validation process. Respondents suggested that because implementation measures were not used directly to guide patient care, the stakes were lower than for other measures (e.g., diagnostic or screening tools), and correspondingly the bar for validation was lower.

Panel members were also asked about how they would choose between different hypothetical implementation measures based on their pragmatic qualities, assuming the hypothetical measures were equally valid. Respondents scored nearly all pragmatic qualities as important in making this decision, though acceptability, ease of completion, cost and language accessibility were rated as the most important qualities that would be considered (Table 2). In follow-up conversations with panel members, nearly all highlighted measure length as a key issue with current implementation measures, raising concerns related to respondent fatigue, assessor fatigue and artificial inflation of internal consistency. Respondents also felt that the results from most currently available measures were difficult to interpret, and that this was holding back their use and applicability. They suggested that the inclusion of quantitative thresholds and other guidance on how to judge what measure scores "mean" would be beneficial.

**Table 2.** Delphi panel pragmatic qualities importance ratings

| | Average score |
|---|---|
| Acceptable (*Do users like the measure?*) | 3.8 |
| Completed with ease (*How hard is the measure to complete?*) | 3.8 |
| Cost (*Is the measure free to use?*) | 3.8 |
| Uses accessible language (*What is the reading level of the measure?*) | 3.8 |
| Appropriate (*Does use of the measure interfere with service implementation?*) | 3.4 |
| Length (*How many items does the measure have?*) | 3.2 |
| Informs clinical or organizational decision-making (*Are the measure findings actionable?*) | 3 |
| Fits organizational activities (*Does the measure map to actual services?*) | 2.8 |
| Assessor burden (training) (*How much training is required to learn how to administer the measure?*) | 2.8 |
| Assessor burden (interpretation) (*Does the measure have clear cut-offs, instructions for handling missing data and generating summary scores?*) | 2.8 |
| Offers relative advantage over existing methods (*Is the measure better than other approaches to assessment of the same construct?*) | 2.4 |

**Table 3.** Investigator survey respondent characteristics (*n* = 28)

| | Total (%) |
|---|---|
| Country of residence (%) | |
| Ethiopia | 1 (3.6) |
| India | 2 (7.1) |
| Lebanon | 1 (3.6) |
| Nepal | 1 (3.6) |
| Netherlands | 1 (3.6) |
| Nigeria | 1 (3.6) |
| Pakistan | 1 (3.6) |
| South Africa | 1 (3.6) |
| Ukraine | 1 (3.6) |
| United Kingdom | 1 (3.6) |
| United States | 17 (60.7) |
| Organization type (%)[a] | |
| University/Academic | 23 (82.1) |
| Non-profit/NGO | 7 (25.0) |
| Government | 1 (3.6) |
| Healthcare setting | 1 (3.6) |
| Role (%)[a] | |
| Academic/research | 27 (96.4) |
| Clinical service delivery | 4 (14.3) |
| Program implementation | 5 (17.9) |
| Policymaking | 1 (3.6) |
| Time working in mental health | |
| 0–4 years | 3 (10.7) |
| 5–14 years | 12 (42.9) |
| 15+ years | 13 (46.4) |
| Mental health implementation studies conducted (mean (SD)) | 2.21 (1.37) |

[a]≥1 response per participant possible.

### Section 2: Validation strategies

Respondents identified a trade-off between the rigor of different validation approaches and their resource-intensiveness (Supplementary Material, Section 2). The two survey-based validation strategies, one using other established measures and the other using subsequent real-world outcomes for validation, were judged to be the most rigorous as well as the most expensive and time-consuming. Respondents rated the two forms of expert elicitation (informal and formal) as moderately or highly feasible and inexpensive, but there was no agreement on the assumed rigor of the results. Translation/back-translation scored consistently and moderately on all dimensions. Respondents disagreed most about the vignette-based strategy; they did not agree on the amount of time and resources required, nor whether it was feasible to develop vignettes that could provide high-confidence results in diverse low-resource settings. One respondent cautioned that developing good vignettes for community mental health programs could be hampered by the fact that these services are often uncommon in low-resource settings, and thus there is no "gold standard" program to which one can refer. Instead, vignettes must use hypothetical examples that take longer to explain and may produce unreliable results.

### Section 3: Package of validation strategies

Translation/back-translation was the most frequently recommended strategy followed by informal expert elicitation. No other strategy was recommended by more than two respondents. Several respondents struggled with the tension between cost and rigor and wondered whether a minimal set of validation strategies might be feasible in most situations but ultimately insufficient for establishing validity. Most respondents suggested using a combination of validation strategies was the most appropriate approach; nearly all respondents argued that strategies should be "fit for purpose" and only as rigorous and complex as necessary. Respondents also debated the most appropriate approach to disseminate guidance on implementation measurement to mental health services researchers across diverse global settings. One respondent argued for the provision of step-by-step guidance, while another cautioned against offering overly prescriptive guidance to LMIC-based investigators.

Complete Delphi panel results are presented in the Supplementary Material.

**Table 4.** Implementation measure usage and adaptation/validation approaches

| | CFIR (*n* = 7) | PSAT (*n* = 5) | AIM, IAM, FIM (*n* = 5) | AMHR/mhIST (*n* = 4) | EBPAS (*n* = 4) | ORIC (*n* = 3) |
|---|---|---|---|---|---|---|
| Implementation phase of application | Pre (*n* = 5); mid (*n* = 4); post (*n* = 2) | Pre (*n* = 2); mid (*n* = 4); post (*n* = 1) | Pre (*n* = 2); mid (*n* = 5); post (*n* = 1) | Pre (*n* = 1); mid (*n* = 2); post (*n* = 2) | Pre (*n* = 4); mid (*n* = 2); post (*n* = 1) | Pre (*n* = 2); mid (*n* = 1) |
| Measure use | Contextual determinant/process (*n* = 6); outcome (*n* = 1) | Contextual determinant/process (*n* = 3); outcome (*n* = 3) | Contextual determinant/process (*n* = 3); outcome (*n* = 2) | Contextual determinant/process (*n* = 1); outcome (*n* = 3) | Contextual determinant/process (*n* = 4) | Contextual determinant/process (*n* = 3) |
| Level of application | Client (*n* = 4); provider (*n* = 6); organizational (*n* = 3); policy (*n* = 2) | Provider (*n* = 5); organizational (*n* = 3) | Client (*n* = 1); provider (*n* = 4); organizational (*n* = 1) | Client (*n* = 4); provider (*n* = 3); organizational (*n* = 2) | Provider (*n* = 4); organizational (*n* = 2) | Provider (*n* = 3); organizational (*n* = 3) |
| Range of sample sizes | 10 (providers, organizational, policy) to 86 (providers) | 5 (providers) to 300 (providers) | 10 (organizational) to 1650 (clients) | 10 (organizational) to 200 (clients) | 5 (providers) to 163 (providers) | 20 (organizational) to 124 (providers) |
| Countries | Nepal, Ethiopia, Nigeria, Vietnam, India, South Africa, Kenya, Mozambique, Zimbabwe | Rwanda, Myanmar (Burma), Thailand, Kenya, Mozambique | Chile, Nepal, South Africa, Mozambique, Ethiopia | Iraq, Myanmar (Burma), Thailand, Ukraine, Rwanda, Sierra Leone, South Africa | Brazil, Ukraine, Kenya, Vietnam | Nepal, Ethiopia, Nigeria |
| Adaption approaches | Translated/back-translated (*n* = 4); stakeholder feedback (*n* = 4); qualitative study (*n* = 3); expert feedback (*n* = 4) | Translated/back-translated (*n* = 4); stakeholder feedback (*n* = 5); expert feedback (*n* = 1) | Translated/back-translated (*n* = 5); stakeholder feedback (*n* = 2); cognitive interviewing (*n* = 1) | Translated/back-translated (*n* = 4); stakeholder feedback (*n* = 3); qualitative study (*n* = 1); expert feedback (*n* = 1) | Translated/back-translated (*n* = 3); stakeholder feedback (*n* = 1); expert feedback (*n* = 2) | Translated/back-translated (*n* = 3); stakeholder feedback (*n* = 1); expert feedback (*n* = 1); cognitive interviewing (*n* = 1) |
| Validation approaches | Pilot-tested, results corroborated with field notes (*n* = 1) | *n* = 0 | *n* = 0 | *n* = 0 | *n* = 0 | *n* = 0 |
| Comments | Challenge choosing constructs for adaptation; need ways to translate scales into qualitative questions | Limited response variability; positive response bias; long; some questions not relevant for providers or clients | Straightforward and short; easy to translate; unsure if valid | Some questions not relevant for clients or providers; too many hypotheticals; limited response variability; positive response bias; long | Abstract | Concept not easy to assess using structured questionnaires; prefer qualitative interviews |

*Note*: Measures reported as used by only one investigator, or used only in a high-income country setting, are not included in Table 4. Responses related to the Acceptability of Intervention, Intervention Appropriateness, and Feasibility of Intervention Measures were collapsed across the scales as there was complete overlap within respondents for these measures. Responses related to the Applied Mental Health Research implementation measures, which include client-, provider-, organizational- and policy-level scales for several implementation outcomes and contextual determinants, were collapsed for the same reason.
AIM, Acceptability of Intervention Measure; AMHR/mhIST, Applied Mental Health Research/Mental Health Implementation Science Tool; CFIR, Consolidated Framework for Implementation Research; EBPAS, Evidence-Based Practice Attitude Scale; FIM, Feasibility of Intervention Measure; IAM, Appropriateness of Intervention Measure; ORIC, Organization Readiness for Implementing Change; PSAT, Program Sustainability Assessment Tool.

### Investigator survey

We invited 107 investigators to participate in the survey or suggest other investigators for participation. Sixty-two investigators responded. We sent survey links to 45 investigators who indicated interest in participation. Thirty-eight investigators started the survey. Table 3 presents the characteristics of the 28 investigators who completed the survey. The majority (61%) were based in the United States, most (82%) were at universities or other academic institutions and almost all (96%) were focused on research as opposed to clinical service delivery or program implementation. Investigators had been involved in a mean of 2.2 implementation studies related to mental health.

Table 4 describes the usage of implementation measures reported by at least two investigators in LMIC settings. The most used implementation measures included the Consolidated Framework for Implementation Research Inner Setting measures (*n* = 7) (Fernandez et al., 2018), the Program Assessment Sustainability Tool (*n* = 5) (Luke et al., 2014) and the Acceptability of Intervention Measure, Intervention Appropriateness Measure and Feasibility of

Intervention Measure (*n* = 5) (Weiner et al., 2017). Measures were most commonly used prior to intervention implementation (*n* = 18) or mid-implementation (*n* = 18) as opposed to post-implementation (*n* = 7) and were most often used to assess contextual determinants of implementation effectiveness (*n* = 20) rather than to assess implementation outcomes (*n* = 9). Providers were the most common group sampled (*n* = 25), followed by clients (*n* = 9). Measures were used in a diverse range of contexts across Latin America, Sub-Saharan Africa, Eastern Europe and South/Southeast Asia. Adaptation approaches were generally limited to translation and back-translation (*n* = 23) and stakeholder feedback (*n* = 16), and only one investigator reported conducting any measure validation prior to use (pilot testing). Limited response variability, positive response bias, measure length and item relevance were the most common challenges reported.

Other measures reported as used by individual investigators included the Implementation Leadership Scale (Aarons et al., 2014), the Theory of Planned Behavior measures (Ajzen, 2011), the Feelings Thermometer (ALWIN, 1997), the Systems Usability

Scale (Lewis, 2018), the Organizational Social Context scale (Glisson et al., 2008), several intervention-specific fidelity scales and several measures developed new for individual studies.

## Discussion

This study sought to improve quantitative implementation measurement in the field of global mental health by generating consensus recommendations on best practices for measure choice and validation and by surveying the field to understand current practice. Our expert panel concluded that pragmatic concerns are key to choosing between measures and validation approaches. They noted that many quantitative implementation measures are lengthy and identified a trade-off between resources and rigor in the various approaches available for adapting and validating implementation measures in diverse global settings. However, they concluded that in many cases, it is sufficient for investigators to establish the face validity of an implementation measure in a new setting through some combination of reviewing the use of that measure in a similar setting, convening an informal expert and stakeholder panel, conducting translation and back-translation and piloting the measure to confirm its dimensionality and internal reliability. Though confirming the predictive validity of a measure by correlating it with subsequent real-world outcomes would be the gold standard for measure validation, panel members felt this was unnecessary prior to using most implementation measures. Survey results suggested that though several implementation measures have been used or are in use in global mental health studies across a variety of levels and study phases, almost none have been formally validated as part of those studies.

Quantitative measures must be reliable, valid and practical to be useful for implementation research or practice, though comprehensive reviews of published implementation measures have noted that the field faces several major issues. These include the poor distribution of quantitative measures across implementation constructs and analytic levels; a lack of measures with strong psychometric qualities; measure synonymy (the same measure items are sometimes used to measure different constructs), homonymy (different measure items are used to measure the same construct) and instability (measure items are often changed with each use) and the reality that many implementation measures exhibit poor pragmatic qualities (Lewis et al., 2018). Nevertheless, a growing number of strong implementation measures do exist: the challenge for investigators in diverse global settings in choosing and adapting these – or developing new ones – and ensuring that they perform well. Notably, the Psychometric and Pragmatic Evidence Rating Scale has been developed through stakeholder consensus to provide clear criteria for measure quality, both to inform measure development and measure choice (Stanick et al., 2019). In addition, domain-specific resources are increasingly available to support investigators in choosing between manifest and latent indicators of implementation process and outcomes, including the HIV Implementation Outcomes Crosswalk (Li et al., 2020).

Several key limitations should be noted. Our expert panel consisted of only seven members, reflecting the relatively small number of individuals with intersecting expertise in global mental health, implementation science and psychometrics. In response, we opted for depth over breadth and sought to reach panel consensus across a wide range of issues related to measure use and validation, rather than for one or two key questions. Our Delphi panel size is considered acceptable for non-statistical analysis (Rowe and Wright, 1999). All panel procedures were carried out during the first 6 months of the COVID-19 pandemic, meaning procedures were remote and sometimes asynchronous. For our survey, we sampled investigators from NIH RePORTER and Grand Challenges Canada; these are two of the most prolific funders of global mental health implementation research, though this approach likely biased our sample toward investigators based in North America. To mitigate this risk, we used snowball sampling to attempt to identify and recruit other investigators that would have been missed with this approach. Our overall response rate was low, which again may reflect the small number of individuals actively using quantitative measures in their global mental health implementation studies; many investigators we contacted declined to participate because they were not using quantitative implementation measures.

Despite these limitations, our findings may directly support the growing field of global mental health implementation research. We have used our results to compile a set of guidance documents for investigators planning to quantitatively measure latent implementation processes and outcomes in diverse global settings. These include a compendium of available measures across implementation constructs and detailed descriptions of common adaptation and validation approaches. This guidance should facilitate rigorous and replicable implementation research in an area of high need, though it is not intended to be prescriptive, and local investigators are encouraged to adapt and apply the guidance only where it is useful. Moving forward, as the quantity and quality of implementation measures designed for use in for diverse global contexts increase (Aldridge et al., 2022), the standards for measure adaptation and validation may also shift. Less emphasis may be placed on establishing measure validity for the sake of scientific rigor, with a corresponding increased emphasis on measure pragmatic qualities and capacity to inform real-world health service delivery.

**Open peer review.** To view the open peer review materials for this article, please visit http://doi.org/10.1017/gmh.2023.63.

**Supplementary material.** The supplementary material for this article can be found at https://doi.org/10.1017/gmh.2023.63.

**Data availability statement.** Study data are not publicly available as they contain information that could compromise the privacy of research participants.

**Acknowledgments.** The study team would like to thank our fantastic panel and the survey participants for their valuable contributions to this research.

**Author contribution.** All listed authors qualify for authorship based on making one or more substantial contributions to the manuscript. C.G.K., K.D., L.A., L.K.M. and E.E.H. contributed to the conceptualization of this study. C.G.K., K.D. and L.A. contributed to the formal analysis. C.G.K. wrote the original draft of the manuscript; K.D., L.A., L.K.M. and E.E.H. contributed to reviewing and editing subsequent drafts of the manuscript. All authors read and approved the final manuscript.

**Financial support.** This study was funded by a grant from the National Institute of Mental Health (#R01MH115495-02S1; PIs: Laura Murray, Izukanji Sikazwe). L.A. was supported by the National Institute of Mental Health T32 training grants in Global Mental Health (#T32MH103210; PI: Judith K. Bass) during study conceptualization and analysis and in Mental Health Services and Systems (#T32MH109436; PIs: Emma Elizabeth McGinty, Elizabeth A. Stuart) during manuscript preparation. E.E.H. was supported by a Mentored Career Development Award from the National Institute of Mental Health (#K01MH116335).

**Competing interest.** None declared.

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
