## [Reviewer Report]

Dear Drs. Bass and Chibanda, 

We wish to submit a new manuscript entitled “Implementation measurement in global mental health: results from a modified Delphi panel and investigator survey” for consideration Cambridge Prisms: Global Mental Health. 

We confirm that this work is original and has not been published elsewhere nor is it currently under consideration for publication elsewhere. We also confirm that we have no competing interests, and that all authors have approved the manuscript for submission.

In this paper, we bring together a panel of experts and build consensus around best practices for implementation measurement in diverse global settings, and survey investigators applying these measures to identify strengths and opportunities in current practice. We hope the results will facilitate novel, rigorous, and replicable implementation research in areas of high need. This manuscript should be of relevance to readers with an interest in implementation science. 

Please address all correspondence concerning this manuscript to ckemp11@jhu.edu. 

Thank you for your consideration of this manuscript. 

Sincerely,

Christopher Kemp, PhD MPH

---

## [Reviewer Report]

A very well-written paper with a concise overview of the background to the study, aims and methods used in this paper. The outcomes and conclusions of the study are particularly helpful to promoting best practice in LMICs undertaking implementation research as this is likely to encourage rather than discourage more work using implementation science. An emphasis on pragmatic considerations as opposed to only focusing on scientific rigour is a useful recommendation.

Minor

Table 1 format is difficult to read

Page 17 (Line 7): “...Our expert panel was consisted of seven members” typo

---

## [Reviewer Report]

The authors report results of a Delphi exercise followed by a survey with global mental health researchers, aiming to improve measurement of implementation outcomes in global mental health.

The authors state that “little to no guidance exists to support investigators in the choice, adaptation, validation, and use of implementation measures”. I agree that there is still a lot of work to do. However, I also believe that research on implementation outcomes has made major progress over the last years, see e.g. [1–5]. These developments are not appropriately considered in this paper. I also missed a clear explanantion of central concepts (pragmatic qualities etc.) and a justification of the need for specific instruments for this field.

Further, I struggled with the methods used. The Delphi panel smaller than recommended [6] and does not really appear representative. Also the vast majority of experts who participated in the survey are from North America. Central output is a list of implementation outcome measures. Results are not put into perspective. It remains unclear how these results will “offer guidance to investigators planning to measure implementation”.

References

1. Hull L, Boulton R, Jones F, Boaz A, Sevdalis N. Defining, conceptualizing and evaluating pragmatic qualities of quantitative instruments measuring implementation determinants and outcomes: a scoping and critical review of the literature and recommendations for future research. Transl Behav Med. 2022;12:1049–64. doi:10.1093/tbm/ibac064.

2. Lengnick-Hall R, Gerke DR, Proctor EK, Bunger AC, Phillips RJ, Martin JK, Swanson JC. Six practical recommendations for improved implementation outcomes reporting. Implement Sci. 2022;17:16. doi:10.1186/s13012-021-01183-3.

3. Mettert K, Lewis C, Dorsey C, Halko H, Weiner B. Measuring implementation outcomes: An updated systematic review of measures’ psychometric properties. Implementation Research and Practice. 2020;1:2633489520936644. doi:10.1177/2633489520936644.

4. Willmeroth T, Wesselborg B, Kuske S. Implementation Outcomes and Indicators as a New Challenge in Health Services Research: A Systematic Scoping Review. Inquiry. 2019;56:46958019861257. doi:10.1177/0046958019861257.

5. Khadjesari Z, Boufkhed S, Vitoratou S, Schatte L, Ziemann A, Daskalopoulou C, et al. Implementation outcome instruments for use in physical healthcare settings: a systematic review. Implement Sci. 2020;15:66. doi:10.1186/s13012-020-01027-6.

6. Okoli C, Pawlowski SD. The Delphi method as a research tool: an example, design considerations and applications. Inform Manag. 2004;42:15–29. doi:10.1016/j.im.2003.11.002.

---

## [Reviewer Report]

This is a well-written paper that will be valuable to the field of global mental health implementation science. It responds to an evident gap in quantitative implementation measurement and methodology in the field. Below I make some minor suggestions that I hope will be valuable:

- In the Abstract, the phrase “establish measure pragmatic qualities” is a bit confusing on first read. I suggest revising to “establish the pragmatic qualities of measures” for clarity.

- My main comment is in regards to the selection and inclusion of panelists and survey participants. It would be helpful in the Methods section to provide more detailed rationale regarding the selection of both. Regarding the expert panel, as acknowledged in the Limitations section, the inclusion of seven experts- a majority of whom are from the United States- seems quite narrow. The authors note that there a limited number of scholars working at the “intersection of implementation science, psychometrics, and global mental health”, though this still seems quite a narrow pool. Some additional rationale would help to provide some clarity.

Similarly, I was surprised that the authors did not include investigators funded by the Global Alliance for Chronic Diseases (GACD)’s Mental Health Programme in their survey. The GACD specifically funds implementation science research in LMICs and is made up a consortium of funding agencies from a diverse array of countries including some LMICs. This would have garnered responses from a more diverse range of investigators and would have provided a less US-centric perspective. The authors do address the limitation of the scope of perspectives, but again more rationale for the selection of the survey sample is warranted.

- Finally, it was hard to view and understand the contents of Table 1 given the formatting. It is likely that the formatting was changed when it was uploaded (it looks like it should have been in landscape but was changed to portrait) but this made it hard to read this important table.

---

## [Reviewer Report]

Thank you for addressing my comments in the revised version of the manuscript. Though my concerns remain regarding the predominance of US-based perspectives captured in this work given that its emphasis is on GMH and LMICs, I do believe it’s a valuable starting point in advancing quantitative implementation measurement. I also found that the limitations have been sufficiently addressed in the manuscript. One point of consideration for further transparency regrading the limitations would be to change “North America” to “United States” when referring to this limitation, as it appears no Canadian or Mexican experts or investigators were included in the study.

---

## [Reviewer Report]

I am pleased to accept your revised paper subject to making the minor change recommended by Reviewer 2.